# Pure Acetylene Semihydrogenation over Ni–Cu Bimetallic Catalysts: Effect of the Cu/Ni Ratio on Catalytic Performance

**DOI:** 10.3390/nano10030509

**Published:** 2020-03-11

**Authors:** Shuzhen Zhou, Lihua Kang, Xuening Zhou, Zhu Xu, Mingyuan Zhu

**Affiliations:** 1College of Chemistry and Chemical Engineering, Yantai University, Yantai 264004, China; Shuzhenzhou@163.com (S.Z.); lhkang@ytu.edu.cn (L.K.); 2School of Chemistry and Chemical Engineering, Shihezi University, Shihezi 832003, China; zhouxuening@stu.shzu.edu.cn (X.Z.); xz20200217@126.com (Z.X.)

**Keywords:** Ni–Cu, calcium carbide acetylene, selective hydrogenation, ethylene

## Abstract

Ethylene is an important chemical raw material and with the increasing consumption of petroleum resources, the production of ethylene through the calcium carbide acetylene route has important research significance. In this work, a series of bimetallic catalysts with different Cu/Ni molar ratios are prepared by co-impregnation method for the hydrogenation of calcium carbide acetylene to ethylene. The introduction of an appropriate amount of Cu effectively inhibits not only the formation of ethane and green oil, thus increasing the selectivity of ethylene, but also the formation of carbon deposits, which improves the stability of the catalyst. The ethylene selectivity of the Ni–Cu bimetallic catalyst increases from 45% to 63% compared with the Ni monometallic counterpart and the acetylene conversion still can reach 100% at the optimal conditions of 250 °C, 8000 mL·g^−1^·h^−1^ and V(H_2_)/V(C_2_H_2_) = 3. X-ray diffraction and transmission electron microscopy confirmed that the metal particles were highly dispersed on the support, High-resolution transmission electron microscopy and H_2_-Temperature programmed reduction proved that there was an interaction between Ni and Cu, combined with X-ray photoelectron spectroscopy and density functional theory calculations results, Cu transferred electrons to Ni changed the Ni electron cloud density in NiCu_x_ catalysts, thus reducing the adsorption of acetylene and ethylene, which is favorable to ethylene selectivity.

## 1. Introduction

As an important petrochemical raw material, ethylene is currently produced by cracking hydrocarbon from petroleum feedstock such as naphtha and ethane [1,2]. With the increasing consumption of petroleum resources, the production of calcium carbide from coal, acetylene from calcium carbide and water, and then hydrogenation to ethylene has emerged as a new alternative approach to produce ethylene, which is more economically feasible in areas rich in coal resources.

Alkynes have stronger adsorbability than alkenes [3] and extensive research on the hydrogenation of acetylene in ethylene-rich streams has shown that the path of direct hydrogenation of acetylene to ethylene is fully feasible [4,5]. Scheme 1 summarizes a plausible reaction route for the acetylene hydrogenation. The adsorbed acetylene can react with the adsorbed hydrogen to form ethylene as the desired product, or hydro-oligomerize to form C_4_H_6_ hydrocarbons. Nondesorbed ethylene can be over-hydrogenated to form ethane [6,7]. In addition, ethylene selectivity also depends on the kinetics. Thus, for k_2_ >> k_4_, the desired ethylene products are obtained. For k_1_/k_−1_ >> k_3_/k_−3_, the adsorption of acetylene may prevent the readsorption of ethylene, thereby preventing the rehydrogenation of ethylene [8]. Unfortunately, if k_1_/k_−1_ >> k_2_/k_−2_, the adsorbed acetylene will hydro-oligomerize to form C_4_H_6_, which is considered as the precursor of green oil and will lead to catalyst deactivation [9]. Therefore, in order to maximize the production of ethylene, ethylene overhydrogenation should be reduced and acetylene polymerization should be inhibited.

The noble metal Pd is widely used in selective hydrogenation of acetylene and other auxiliary metals such as Ag, Cu, Au, and Zn are usually added to change the geometric and electronic effects of Pd, which allows regulating the adsorption of reactants and products on Pd and achieve a good catalytic effect [10,11]. Unfortunately, in the calcium carbide acetylene hydrogenation reaction, the acetylene concentration is so high that a high number of active sites are required, concomitantly increasing the catalyst cost. Therefore, the development of a cost-effective catalyst suitable for this reaction system with satisfactory activity is highly desirable. Among the nonprecious metals, Ni is considered the most active in the hydrogenation reaction, reaching a high acetylene conversion, albeit with a poor ethylene selectivity [12,13]. Compared with hydrogen, Ni has a stronger adsorption for acetylene, which facilitates the polymerization of acetylene to form green oil once it is adsorbed on the Ni surface [13]. Therefore, to enhance ethylene selectivity on Ni-based catalysts, the adsorption of ethylene and acetylene should be reduced to avoid over-hydrogenation of ethylene and polymerization to form green oil, respectively. To achieve this, a second metal can be introduced to change the electron cloud density on the Ni surface, thus changing the adsorption energy of Ni for acetylene and ethylene. In this context, the addition of metals such as Zn, Ga, and Ag on Ni catalysts has been investigated. Spanjers et al. reported that Ni/Zn intermetallic compounds significantly reduce the adsorption of acetylene compared to their Ni monometallic counterpart, thereby reducing the formation of oligomeric species and increasing ethylene selectivity [13]. Wang et al. demonstrated that the ethylene selectivity of Ni/SiO_2_ could be enhanced from 55% to 75–81% in Ni_5_Ga/SiO_2_ after adding Ga, due to the Ga-induced change in the geometric and electronic effects of Ni atoms inhibiting excessive hydrogenation and polymerization [14]. Chen et al. reported that inserting In into a Ni/SiO_2_ catalyst geometrically isolated Ni atoms and caused electron transfer from In to Ni, which reduced ethylene adsorption and inhibited acetylene polymerization, thereby improving ethylene selectivity from 45% to 60% [15]. Pei et al. reported the introduction of Ag in the Ni/SiO_2_ catalyst, and the interaction between Ni and Ag improved the performance of the catalyst [16]. In addition, Yang et al. demonstrated by density functional theory (DFT) calculations that the addition of Cu to a Ni catalyst can reduce the adsorption energy of acetylene and increase the catalytic activity [17]. Meanwhile, Liu et al. reported the preparation of a NiCu nanoalloy catalyst using a layered double hydroxide for the ethylene-rich acetylene removal reaction [18]. Compared with the Ni monometallic catalyst, although the acetylene conversion decreased from 100% to 85%, the ethylene selectivity increased from 22.1% to 70.2%, and the catalyst stability was also improved, but they did not consider the effect of Cu content on the catalytic performance, and also ignored the green oil selectivity.

To gain more insight into the influence of adding Cu to Ni catalysts for pure acetylene hydrogenation to ethylene, we herein examine the performance of NiCu_x_ catalysts considering green oil selectivity. We prepared a series of catalysts with different Cu/Ni molar ratios by coimpregnation, and used amino-modified MCM-41 as a support to improve the dispersibility of metal particles. Characterization techniques such as X-ray diffraction (XRD) and transmission electron microscopy (TEM) were used to analyze the geometric structure of the catalyst, and the interaction between Ni and Cu was confirmed by High resolution transmission electron microscope (HRTEM), H_2_ temperature-programmed reduction (H_2_-TPR), and X-ray photoelectron spectroscopy (XPS) characterization. In addition, we calculated using DFT the adsorption energy of the catalyst for acetylene and ethylene before and after the addition of Cu. Furthermore, the reason for the different stability of the catalysts was investigated by thermogravimetric analysis (TGA).

## 2. Experimental Sections

### 2.1. Materials 

MCM-41 (surface area: 1124.0 m²/g) was purchased from Nanjing Xianfeng Nano Material Technology Co., Ltd. (Nanjing, China). Nickel nitrate hexahydrate (Ni(NO_3_)_2_·6H_2_O) (AR, 98%) and copper nitrate trihydrate (Cu(NO_3_)_2_·3H_2_O) (AR, 99%) were purchased from Shanghai Macklin Biochemical Technology Co., Ltd. (Shanghai, China). 3-aminopropyltriethoxysilane (APTES) (AR, 98%) was purchased from Aladdin Industrial Co., Ltd. (Shanghai, China).

### 2.2. Catalyst Preparation

MCM-41 was subjected to the following treatment, which was reported in our previous work [19,20]: 120 mL of toluene was placed in a 250 mL three-necked flask, 2 g of MCM-41 was added under stirring, and then 4.5 mL of 3-aminopropyltriethoxysilane (APTES) was added dropwise under N_2_ protection. After 12 h, it was suction filtered, washed with deionized water, and then dried at 100 °C.

A series of catalysts were prepared by coimpregnation method. A certain amount of Nickel nitrate and Cupric nitrate was dissolved in 30 mL of deionized water, and 2 g of MCM-41 was added under stirring, ultrasonically dispersed for 10 min, stirred for 24 h, and then evaporated to dryness at 70 °C obtained solid powder. The dried sample was calcined in a muffle furnace at a rate of 5 °C/min to 400 °C for 4 h, and then reduced in a tube furnace under 10 vol% H_2_/Ar atmosphere at a rate of 5 °C/min to 500 °C for 4 h to obtain the NiCu_x_/MCM-41 catalysts (x represents the molar ratio of Cu to Ni). In this study, the Ni load was fixed at 1 wt%, and the Cu content was adjusted to achieve the Cu/Ni molar ratio of 0.05–2. For comparison, 1%Ni/MCM-41 and 1%Cu/MCM-41 catalysts were prepared following the same method.

### 2.3. Catalyst Characterization

The X-ray diffraction (XRD) pattern was obtained by measuring the catalyst on the Bruker D8 Advance X-ray diffractometer (Billerica, MA, USA) with Cu-Kα irradiation (λ = 1.5406 Å) as an X-ray source. We heated 20 mg catalyst before reduction to 900 °C at a heating rate of 10 °C/min on the Quantachrome Instruments automated chemisorption analyzer (Boynton Beach, FL, USA) to obtain the temperature-programmed reduction (TPR) pattern. Transmission electron microscopy (TEM) images were obtained with a Tecnai F30 field emission transmission electron microscope (Hillsboro, OR, USA) operating at 300 KV at room temperature. The X-ray photoelectron spectroscopy (XPS) data of Ni in the catalyst were obtained on a Thermo Fisher Scientific ESCALAB 250Xi X-ray photoelectron spectroscopy analyzer (Waltham, MA, USA). A Thermo-ICAP 6300 plasma emission spectrometer (Bremen, Germany) was used to obtain the actual Ni content of the catalyst. Before the test, the 0.1 g catalyst was completely dissolved into a clarified solution with hydrofluoric acid and aqua regia, and then diluted to a 50 mL volumetric bottle. Thermogravimetric (TG) analysis was performed to analyze the carbon deposit on the catalyst. The catalysts were analyzed on a Netzsch synchronous thermal analyzer (Selb, Germany) from room temperature to 900 °C with a 10 °C/min heating rate.

### 2.4. Catalytic Performance Test

The pure acetylene hydrogenation to ethylene reaction was performed continuously on a fixed bed reactor under 0.1 MPa. The catalyst (0.1 g) was placed into the reactor (a stainless steel tube with a diameter of 10 mm) and heated to 150 °C in H_2_ (80 mL/min) to provide the initial temperature for the reaction. After the temperature was stabilized, the H_2_ flow rate was adjusted to a certain value, a certain amount of acetylene was introduced into the reaction system, and the temperature was maintained at the specified reaction temperature (no other gas participates in the reaction). The raw materials, acetylene and hydrogen, were of more than 99.99 vol% purity. The gaseous products were analyzed online using a Shimadzu GC-2014C gas chromatograph (Shimadzu, Kyoto, Japan) and detected by a thermal conductivity detector. 

Acetylene conversion (*X*) and product selectivity (*S*) were calculated using the following formulas:$$X_{C_{2}H_{2}}=\frac{n_{C_{2}H_{2}\left(\mathrm{inlet}\right)}-n_{C_{2}H_{2}\left(\mathrm{outlet}\right)}}{n_{C_{2}H_{2}\left(\mathrm{inlet}\right)}}$$
$$S_{C_{2}H_{4}}=\frac{n_{C_{2}H_{4}\left(\mathrm{outlet}\right)}}{n_{C_{2}H_{2}\left(\mathrm{inlet}\right)}-n_{C_{2}H_{2}\left(\mathrm{outlet}\right)}}$$
$$S_{C_{2}H_{6}}=\frac{n_{C_{2}H_{6}\left(\mathrm{outlet}\right)}}{n_{C_{2}H_{2}\left(\mathrm{inlet}\right)}-n_{C_{2}H_{2}\left(\mathrm{outlet}\right)}}$$
$$S_{green\;oil}=1-S_{C_{2}H_{4}}-S_{C_{2}H_{6}}$$

In the above formula, green oil consists of C_4+_ hydrocarbons and carbon deposition based on carbon balance calculation.

Average turnover frequency (*TOF*) is an important parameter to evaluate the activity of catalyst, which was calculated as follows.
$$TOF_{C_{2}H_{2}}=\frac{X\times n_{C_{2}H_{2}}}{n_{\left(\mathrm{Ni}+\mathrm{Cu}\right)}\times D}$$

X = the acetylene conversion, $n_{C_{2}H_{2}}$ = the molar rate of acetylene per second, $n_{\left(\mathrm{Ni}+\mathrm{Cu}\right)}$ = the mole of Ni and Cu in the catalyst, *D* = dispersion calculated by particle size.

The stability test was conducted using fresh catalysts for continuous reaction until the catalyst is deactivated under optimal reaction conditions of 250 °C, 8000 mL·g^−1^·h^−1^ and V(H_2_)/V(C_2_H_2_) = 3. The catalyst testing process is consistent with the catalyst performance test process described above.

## 3. Results and Discussion

### 3.1. Catalyst Activity and Characterization

To determine the actual metal content in the catalyst and the molar ratio of Cu to Ni, we subjected the reduced catalyst to inductive coupled plasma (ICP) characterization. The results are shown in Table 1. The actual loading of Ni is close to the theoretical value, indicating that Ni was successfully loaded onto MCM-41. In contrast, the actual Cu loading was lower than the theoretical value, indicating that Cu was harder to load onto the support than Ni. As a result, the actual molar ratio of Cu to Ni was slightly lower than the theoretical value. Appendix A shows the XRD patterns of catalysts after reduction at 500 °C. All samples had a broad and distinct peak between 20° and 40°, which can be assigned to the characteristic peak of SiO_2_ in the MCM-41 support [21]. However, no characteristic peaks of Ni or Cu were observed in the samples, which might be due to the small size of the metal nanoparticles or to low metal loading [16]. 

We characterized the particle size of the NiCu_x_/MCM-41 (*X* = 0, 0.125, 0.5) catalysts by TEM (Figure 1). The metal particles are more uniformly dispersed on the MCM-41 surface, and the average particles of the Ni/MCM-41, NiCu_0.125_/MCM-41, and NiCu_0.5_/MCM-41 catalysts after reduction were calculated to be 3.99, 4.94, and 4.85 nm, respectively. Compared with the Ni/MCM-41 catalyst, the growth of the catalyst particles after Cu addition is small, but the metal particle size is still uniformly, and no larger particles or agglomeration appear. These results are consistent with the XRD results.

We explored the effect of having different Cu/Ni molar ratio on the catalyst activity. The results are shown in Figure 2. Under the same reaction conditions, the acetylene conversion for the Ni/MCM-41 catalyst remained 100% within 9 h. The selectivity of ethylene, ethane, and green oil were 45%, 10%, and 45%, respectively. For the NiCu_x_/MCM-41catalyst, with the increase of Cu content, the initial acetylene conversion in all cases reached 100%, but the stability gradually decreased, with the catalyst having the molar ratio Cu/Ni ≤ 0.25, being relatively stable during the test time. The ethylene selectivity first increased and then decreased, and the catalyst with Cu/Ni = 0.125 reached the optimal level. The ethane selectivity remained below 3%, and the green oil selectivity decreased first and then increased. This phenomenon is consistent with previous findings [15]. Finally, NiCu_0.125_/MCM-41 exhibited the best hydrogenation activity. During the test period, the acetylene conversion remained 100%, the ethylene selectivity was 63%, the ethane selectivity was only approximately 2%, and the green oil selectivity was approximately 35%. Compared with the ethylene selectivity of the Ni/MCM-41 catalyst, that of the NiCu_0.125_/MCM-41 catalyst increased by 18%, mainly because the addition of Cu reduced the formation of ethane and green oil. It is worth noting that the catalyst stability was negatively correlated with the green oil selectivity; the catalyst deactivated faster as the green oil selectivity increased, which indicates that the accumulation of green oil with reaction time was detrimental to the stability of catalyst. The TOF value of Ni/MCM-41, NiCu_0.125_/MCM-41, and NiCu_0.5_/MCM-41 catalysts were calculated as 1.87, 2.04 and 1.79, respectively (Figure 3). Obviously, compared with the other two catalysts, NiCu_0.125_/MCM-41 had the largest TOF value, indicating that the catalyst had excellent catalytic activity.

To further understand the geometric structure of Ni–Cu in bimetallic catalysts, the lattice sizes of metal particles in the NiCu_0.125_/MCM-41 and NiCu_0.5_/MCM-41 catalysts were characterized by HRTEM (Figure 4). It was found that the lattice fringe of individual particles in the NiCu_0.125_/MCM-41 and NiCu_0.5_/MCM-41 catalysts was 0.206 and 0.207 nm, respectively, both between those of Ni[111] (0.203 nm) and Cu[111] (0.209 nm), indicating that an intermetallic compound was formed between Ni and Cu [14,21].

Figure 5A shows the H_2_-TPR curves of Ni/MCM-41 and Cu/MCM-41 and the bimetallic NiCu_x_/MCM-41 catalysts. Ni/MCM-41 showed a wide peak at 380–630 °C, which is attributed to the reduction of NiO particles having strong interactions with the MCM-41 support [22,23]. Meanwhile, Cu/MCM-41 exhibited two distinct peaks at 227 °C and 310 °C due to the reduction of Cu^2+^ and Cu^1+^, respectively [24,25]. The series of Ni–Cu/MCM-41 catalysts displayed two main reduction peaks, a low temperature reduction peak attributed to the reduction of copper oxide and a high temperature reduction peak that can be assigned to the reduction of nickel oxide. The shift of the reduction peak to low temperature indicates the occurrence of an interaction between Ni and Cu. As the Cu content increased, both reduction peaks shifted toward the low temperature direction; the decrease in the reduction temperature of nickel oxide is attributed to the fact that Cu species participate in the reduction of Ni species [26,27]. Moreover, as the Cu content increased, the reduction peak of nickel oxide became wider, further indicating that there was a strong interaction between Ni and Cu species, which led to the formation of a Ni–Cu intermetallic compound [15]. This result is consistent with the HRTEM result.

Figure 5B display the XPS spectra in the Ni2p_3/2_ region for Ni/MCM-41, NiCu_0.125_/MCM-41, and NiCu_0.5_/MCM-41. In the spectrum of Ni/MCM-41, the electron binding energy at 852.86 eV corresponds to zero-valent metal nickel, and the binding energy at 856.51 eV and 862.41 eV can be assigned to nickel oxide with the Ni2p_3/2_ and its satellite peak, respectively [28,29,30]. After the addition of the Cu species, Ni2p_3/2_ shifted to a low binding energy of 852.76 eV for NiCu_0.125_/MCM-41 and 852.62 eV for NiCu_0.5_/MCM-41. Figure 5C display the XPS spectra in the Cu2p_3/2_ region for NiCu_0.125_/MCM-41, and NiCu_0.5_/MCM-41. The low electron binding energy and high electron binding energy are attributed to Cu^0^ and Cu^2+^, respectively [18] and Cu2p_3/2_ shifts to a high binding energy with the increase of Cu content. These changes in the binding energies of Ni and Cu indicate that the electron transfer from Cu to Ni changed the electron cloud density of the Ni surface, and is consistent with the formation of a Ni–Cu intermetallic compound [27,31]. This result is in agreement with those of the HRTEM and TPR analysis.

Moreover, the adsorption energy of acetylene, ethylene, and hydrogen on the metal clusters was determined by DFT calculation. All simulations were performed using the Guassian09 software package (Gaussian Inc.: Wallingford, CT, USA) [32] with the hybrid B3LYP density functional method [33]. The 6-31G basis set was applied for nonmetals and the LANL2DZ pseudopotential basis set for metals. According to the approximate actual molar ratio of Cu to Ni obtained from the experimental results, we constructed Ni–Cu metal clusters as the initial model for the evaluation of the adsorption. The model depicted in Appendix A showed the most stable optimized structure for each catalyst. After that, we studied the adsorption of acetylene and ethylene molecules by five different catalysts, and obtained the optimal adsorption configuration. The optimized adsorption structure can be seen in Appendix A, and the adsorption energy is shown in Figure 6. From the comparison of the adsorption energy, it can be extracted that the acetylene adsorption energy was higher than the hydrogen adsorption energy for all the catalysts, which indicates that acetylene was more easily adsorbed at the active site during the adsorption process. Although the adsorption of acetylene on Ni clusters was significantly reduced after the addition of Cu, there was no significant difference in the adsorption energy of acetylene for the metal clusters with different molar ratios of Cu to Ni. In any case, the decrease in the acetylene adsorption energy reduces the formation of polymer. In addition, the hydrogen adsorption on NiCu_x_ metal clusters was lower than that on Ni metal clusters, which was advantageous for reducing excessive hydrogenation of ethylene. At the same time, the addition of Cu also reduced the adsorption of ethylene, which accelerated the desorption of ethylene product from the active site, resulting in higher ethylene selectivity. Interestingly, the adsorption energy of metal clusters to ethylene decreased with the decrease of Cu content, which is consistent with the experimental results that indicate that the ethylene selectivity increases with the decrease of Cu content in the catalyst. Overall, the formation of Ni–Cu intermetallic compounds changes the adsorption of Ni to acetylene, ethylene, and hydrogen, thus improving the ethylene selectivity.

To clarify the reason for the different stability of the catalysts having different Cu/Ni molar ratios, we conducted thermogravimetric tests on the catalysts after reaction for 9 h. The results are shown in Figure 7. Two significant weightlessness peaks appeared for all catalysts. The weightlessness at low temperature (210–390 °C) can be attributed to the combustion of light hydrocarbons adsorbed in the catalyst pores or on the catalyst surface [34,35], whereas the weightlessness at high temperature (>390 °C) is attributable to the combustion of heavy hydrocarbons or coke deposited on the catalyst surface [36,37]. Note that compared with the Ni catalyst, the weightlessness peak significantly moved toward the high temperature region after adding Cu, which indicates that the green oil deposits changed to hydrocarbons heavier than those deposited on the Ni catalyst surface. As can be seen from Figure 7, except NiCu_0.125_/MCM-41, the mass loss was lower than that of the Ni catalyst and increased with the Cu content. This indicates that the addition of an appropriate amount of Cu reduced carbon deposition, whereas the excess amount of Cu had the opposite effect, leading to catalyst deactivation as the Cu content increased. Interestingly, the weight loss of different Cu/Ni molar ratios after the reaction can be positively correlated with the selectivity of green oil in the reaction, indicating that the carbon deposition was mainly caused by green oil.

### 3.2. Optimization of Reaction Conditions

Considering the above study, which evinced the good stability and optimal selectivity of NiCu_0.125_/MCM-41, this catalyst was selected to explore the effects of the reaction conditions (i.e., temperature, acetylene space velocity, acetylene to hydrogen ratio) on the conversion and product selectivity. Appendix A show the effect of reaction temperature on acetylene conversion and ethylene selectivity. The acetylene conversion was almost above 97% in the 150–300 °C temperature range. The catalyst stability increased significantly with temperature (150–250 °C), which may be due to volatilization of part of the hydrocarbons attached to the catalyst surface at high temperature. However, at 300 °C, the stability of the catalyst decreased slightly, most likely due to catalyst sintering. By contrast, the effect of temperature on ethylene selectivity was negligible.

The effect of acetylene space velocity on acetylene conversion and ethylene selectivity is evaluated in Appendix A, which show that the catalyst stability decreased gradually with the increase of acetylene space velocity, and the ethylene selectivity increased significantly first and then remained basically unchanged. This is mainly caused by the increase in the accumulation of green oil on the catalyst surface at high space velocity, whereas the over-hydrogenation of ethylene into ethane is hampered at short residence times [38,39]. Appendix A show the effect of the acetylene to hydrogen ratio on acetylene conversion and ethylene selectivity. For hydrogen to acetylene ratios in the range of 2–6, the acetylene conversion was above 98% and relatively stable as the ratio increased; the ethylene selectivity increased first and then remained virtually unaltered. This result is consistent with the literature report [40], and is due to the competitive adsorption of acetylene, hydrogen, and ethylene on the catalyst in this reaction. With the increase of the hydrogen to acetylene ratio from 2 to 3, the hydrogen coverage on the catalyst surface was enhanced, which weakened the adsorption of acetylene, reduced the probability of acetylene polymerization, and increased ethylene selectivity. At higher hydrogen concentrations (hydrogen to acetylene ratio > 3), the hydrogen adsorbed at the active site caused further hydrogenation of the intermediate ethylene to ethane [9,39]. Finally, the optimal reaction conditions were T = 250 °C, GHSV = 8000 mL·g^−1^·h^−1^, and hydrogen to acetylene ratio = 3. In these conditions, the acetylene conversion could be kept at 100% within 9 h, and the ethylene selectivity reached 63.6%.

### 3.3. Stability Test

To investigate the effect of Cu on catalysts stability, stability tests were conducted for NiCu_0.125_/MCM-41 and Ni/MCM-41. It can be seen from the experimental results depicted in Figure 8 that the stability improved significantly after adding Cu and the TOF value of NiCu_0.125_/MCM-41 higher than that of Ni/MCM-41. The acetylene conversion of NiCu_0.125_/MCM-41 maintains over 99% for 42 h, whereas the same value of acetylene conversion was kept for 24 h for the Ni/MCM-41 catalyst. According to the TG result displayed in Figure 7B, the weight loss of NiCu_0.125_/MCM-41 was lower than that of Ni/MCM-41 after the reaction, indicating that the decrease of carbon deposition on the catalyst surface was the reason for the good stability of the NiCu_0.125_/MCM-41 catalyst.

## 4. Conclusions

A series of Ni–Cu bimetallic catalysts supported by amino-modified MCM-41 were prepared by coimpregnation. Compared with the Ni/MCM-41 catalyst, the ethylene selectivity of the bimetallic NiCu_x_/MCM-41 catalyst was significantly improved, mainly attribute to the interaction between Ni and Cu reducing the adsorption energy of acetylene and ethylene, thereby hindering acetylene polymerization and ethylene hydrogenation. In addition, with the decrease of Cu content, the stability of the catalyst gradually increased, which was mainly due to the decrease of carbon deposition. In general, the NiCu_0.125_/MCM-41 catalyst exhibited better ethylene selectivity and stability than the Ni/MCM-41 catalyst. Our results provide useful information for the design of Ni-based catalysts for the selective hydrogenation of pure acetylene to ethylene.

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
