# Peer review of "Pure Acetylene Semihydrogenation over Ni–Cu Bimetallic Catalysts: Effect of the Cu/Ni Ratio on Catalytic Performance"

_nanomaterials, 2020, doi:10.3390/nano10030509_

Round 1

Reviewer 1 Report

The paper describes the preparation & characterization of several bimetallic catalysts with different Cu/Ni ratios by coimpregnation method. The paper requires major modifications as many essential data are omitted. Especially the XPS spectra for Cu are missing. While the shift in the Ni band is clearly demonstrated, it would be good to see an opposite shift in the Cu lines. Strictly speaking these data are needed to confirm true bimetallic character of the catalysts obtained. Also it should be mentioned how the samples were pretreated before these experiments. The long term stability data should be provided. The data should be obtained at several different temperatures. The catalytic activity data should be compared with literature data.

Other comments:

  1. The abstract should be made more informative. Reaction conditions and the main outcomes should be mentioned explicitly.
  2. The accuracy of all numbers should be checked. Line 95: 1124.0137 m²/g. This does not look realistic.
  3. The purity of all chemicals should be mentioned in the experimental section.
  4. The heating rates during catalyst preparation should be mentioned.
  5. The percentage should be explained whether this is wt., vol. or mol.%. For example, line 110: Ni load was fixed at 1%, line 109: 10% H2/Ar and so on.
  6. Figure 1. The XRD patents look identical. It is not clear why this Figure was presented? The MCM-41 pattern is observed in all samples.
  7. The absence of mass transfer limitations in the reactor should be verified and discussed prior to discussion of kinetic and adsorption parameters.
  8. The particle size in the reactor and the length of the catalyst bed should be mentioned. How the catalyst temperature was measured?
  9. The N2 chemisorption data should be presented before and after catalyst deposition and after catalyst regeneration.
  10. How the catalysts were regenerated at the end of experimental runs (Figure 7)?\
  11. Figure 7. The amount of catalyst is far too much to observe any difference. The experiment needs to be repeated with a smaller catalyst amount and the second run after catalyst regeneration should be presented.
  12. The catalyst dispersion should be calculated from the particle size.
  13. The data should be presented in terms of TOF in order to compare with available literature data.
  14. The samples were tested at a single temperature. This is not enough as several temperature are required to calculate the kinetic parameters.

Reviewer 2 Report

The manuscript 'Pure acetylene semihydrogenation over Ni–Cu bimetallic catalysts: Effect of the Cu/Ni ratio on catalytic performance' is interesting and should be accepted for publication after some corrections/modifications. see suggestions below:
1- the introduction emphasises on the removal of acetylene from ethene feeds, which is needed to prevent deactivation of the polymerisation reaction. The present work is on selective hydrogenation of acetylene to ethene that is a new way of making ethene: it should be emphasised that this is the preferred reaction when acetylene is the main reagent due to triple bond adsorption strength compared with a double bond. The authors must change the introduction focus to the target reaction, not a process that has little to do with their work.

2- the statement 'Currently, research on the hydrogenation of acetylene mainly focuses on the removal of 0.1%– 1% acetylene impurities from an ethylene-rich stream based on the petroleum route' should be removed because once again these are completely different reactions. One is to clean a feed the other is to produce ethene from acetylene. While there are commonalities, the problems in both reactions are different

3- there isn't any description of how the authors made the ICP measurements.

4- Figure 1 (wrongly identified in the text) shows no relevant information and should be moved to SI.

5- the catalytic performance is performed in flow-mode and thus data must be also plotted as activity as well.

6- a graph must be added with the overall carbon balance.

7 - figure 7 should be complemented with activity plot.

8- why does the catalytic drops in catalytic activity after 45h on stream? This is an important question because for both Ni and bimetallic Ni-Cu the catalysts are stable for a long time and suddenly they drop significantly in catalytic activity. Therefore what is happening for this catastrophic drop in catalytic activity.

9- post-reaction analysis of the catalysts must be performed. TPO and TPR ara least to see what is going on.

10- authors must compare activity with Pd and Pd-Ag catalysts. Since there are few works on this topic, the authors must provide a frame of reference. Mentioning Pd catalysts from a different reaction isn't good enough.

Round 2

Reviewer 1 Report

The authors provided additional data in the revised version.

However in the answer to Q14 it is stated "The optimum catalyst was tested at different temperatures to investigate the effect of temperature on the reaction"

I do not see which Figure or Table supports this statement. Steady state activity data should be presented in a Figure or in a Table. The corresponding reaction conditions should be provided in the Figure caption.

Author Response

The authors provided additional data in the revised version.

Question: However in the answer to Q14 it is stated "The optimum catalyst was tested at different temperatures to investigate the effect of temperature on the reaction"

I do not see which Figure or Table supports this statement. Steady state activity data should be presented in a Figure or in a Table. The corresponding reaction conditions should be provided in the Figure caption.

Response: The effect of temperature on the reaction was shown in Fig. S4A and Fig. S4B in the supplementary files.